# Distribution of Carbon and Nitrogen as Indictors of Environmental Significance in Coastal Sediments of Weizhou Island, Beibu Gulf

**Zhiyi Tang [2], Chao Cao [1,3,*], Kunxian Tang [1,3], Hongshuai Qi [1,3], Yuanmin Sun [1,3] and Jiangbo Yang [4]**

[1]  Third Institute of Oceanography, MNR, Xiamen 361005, China; tangkunxian@tio.org.cn (K.T.);
    qihongshuai@tio.org.cn (H.Q.); sunyuanmin@tio.org.cn (Y.S.)
[2]  Hubei Key Laboratory of Marine Geological Resources, China University of Geosciences,
    Wuhan 430074, China; a1269167255@126.com
[3]  Fujian Provincial Key Laboratory of Marine Ecological Conservation and Restoration, Xiamen 361005, China
[4]  College of Marine Science and Technology, China University of Geosciences, Wuhan 430074, China;
    18872331994@163.com
*  Correspondence: caochao@tio.org.cn; Tel.: +86-592-2195306

**Abstract:** Carbon and nitrogen contents and their isotopic components, and AMS (Accelerator Mass Spectrometry) radiocarbon dating ages, were measured for 57 coastal sediments from Weizhou Island to analyze the distribution of total inorganic carbon (TIC) and its carbon and oxygen isotopic components ($\delta^{13}C_{carb}$ and $\delta^{18}O_{carb}$), total organic carbon (TOC) and total nitrogen (TN) contents and their stable isotopic components ($\delta^{13}C_{TOC}$ and $\delta^{15}N_{TN}$), and their environmental significance. The results showed that the oldest age of coastal sediments on Weizhou Island was 2750 cal. a BP (before present), and the average TIC contents of cores A1, A2, B1, C1, and D1 in the intertidal zone were all greater than 5%, where $\delta^{13}C_{carb}$ and $\delta^{18}O_{carb}$ were enriched, whereas the TIC contents in cores A3, C2, and D2 of the supra-tidal zone were low, where $\delta^{13}C_{carb}$ and $\delta^{18}O_{carb}$ were depleted. Moreover, TIC decreased sharply, 4.95% on average, to close to zero from the estuary to the upstream region in the C1-C2 section. The average C/N ratio was 7.02, and $\delta^{13}C_{TOC}$ and $\delta^{15}N_{TN}$ were between −14.96‰ and −27.26‰ and −14.38‰ and 4.12‰, respectively. These measurements indicate that the TIC in coastal sediments mainly came from seawater. Cores A1, A2, and B1 in the northern intertidal zone exhibited organic terrestrial signals because of $C_3$ and $C_4$ plant inputs, which indicates that the important source on the northern coast of Weizhou Island came from island land but followed the decrease in $C_3$ plants. The lacustrine facies deposits were mainly distributed in the upper reaches of the river, the northern coastline was advancing toward the sea, and part of the southwestern coastal sediments rapidly accumulated on the shore under the influence of a storm surge. The relative sea level of the Weizhou Island area has continuously declined at a rate of approximately 2.07 mm/a, using beach rock as a marker, since the Holocene.

**Keywords:** inorganic carbon; organic carbon and nitrogen; carbon and nitrogen isotopes; coastal sediments; environmental significance

---

## 1. Introduction

Continental margin sediments contain more than 90% of the carbon buried in the ocean, and the estuary coastal zone, as a sensitive exchange zone for sea–land carbon pools, effectively indicates that the global carbon cycle process is driven by sea level changes [1,2]. However, as an open system,

the biogeochemical signals of coasts are complex due to the influence of multiple natural factors (such as global coastline and paleoclimate changes) and the interference of human activities [3–7].

The most important form of inorganic carbon in marine sediments is carbonate, whose precipitation or dissolution is controlled by the overlying water and pore water ion concentration, pH, temperature, $CO_2$ diffusion rate, and biological disturbance [8,9]. Because total inorganic carbon (TIC) mainly comes from water evaporation, lighter $^{12}C$ and $^{16}O$ are preferentially gathered in the reactants under the isotope dynamic fractionation mechanism. With the diffusion of $CO_2$ into the atmosphere, only heavier $^{13}C$ and $^{18}O$ remain [10]. Although coastal organic matter is complex due to its origin of marine organic matter and terrestrial plant detritus [11], the ratio of total organic carbon to total nitrogen (TOC to TN) and their stable isotope components ($\delta^{13}C_{TOC}$ and $\delta^{15}N_{TN}$) under different sources can distinguish the sources of organic matter, which is helpful for reconstructing the evolution of the environment and the process of human activities [12–17]. Therefore, TIC, organic carbon and nitrogen, and their stable isotopic components are not only the main components of biogeochemistry in coastal sediments but also the main response indicators of environmental evolution [18]. Gao et al. [19] noted that the contribution rate of terrestrial organic matter could be obtained by calculating the $\delta^{13}C_{TOC}$ component. Bejugam and Nayak [20] believed that calcium carbonate was related to the particle size of sediments and the concentration of carbonate ions in seawater, and the C/N ratio was associated with the source of organic matter in rivers, such as the Mahanadi River. Zhan et al. [21] inferred that the decrease in C/N and the increase in $\delta^{13}C_{TOC}$ in the sediments of the Yangtze River Estuary at 9 cal. Ka BP (before present) were related to the decrease in terrestrial source input caused by abrupt sea level rise.

Weizhou Island is located in the central part of the Beibu Gulf in the South China Sea (Figure 1) and has a tropical monsoon climate zone with an average annual temperature of 22.6 °C and rainfall of 1380.2 mm. Under the control of the monsoon, the south and southwest winds and waves are strong, resulting in significant erosion in the south and west regions of Weizhou Island, whereas the wave-shadow areas in the northern part of the island accumulate under favorable coastal currents and coral reef unloading. This process results in the transgression of the southern coastline but the regression of the northern coastline [22,23].

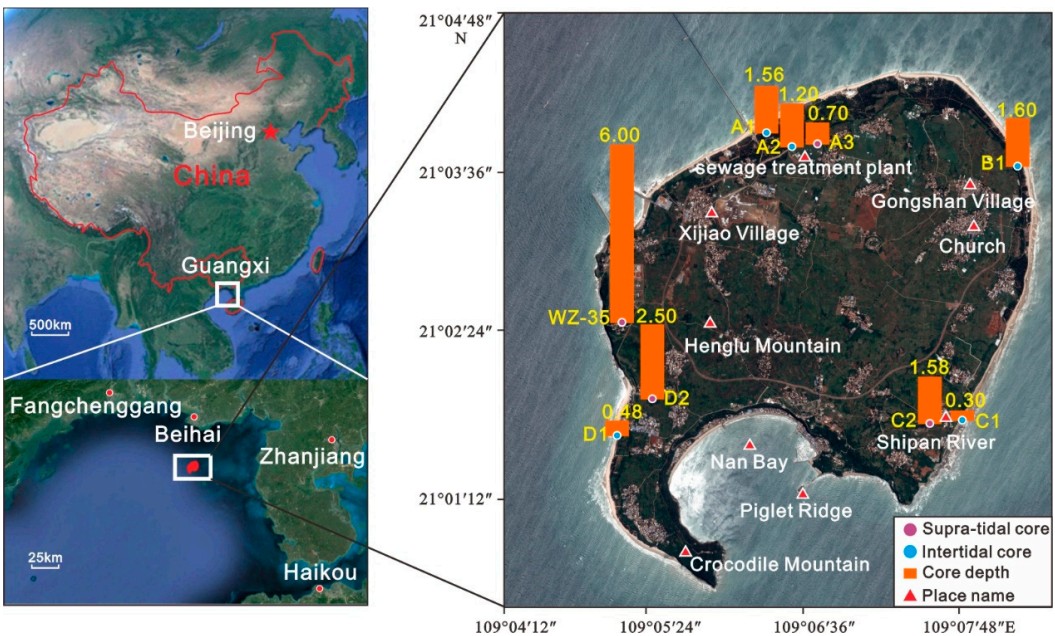

**Figure 1.** Map of study area showing the core location on Weizhou Island. Country and place names are expressed by red and white color, respectively, and core name and depth in yellow with orange columns. In addition, supra-tidal and intertidal cores are displayed with purple and blue circles, and place locations with red triangles.

As a young island that originated from a volcanic eruption, Weizhou Island has experienced tectonic uplift, weathering, and erosion since the Holocene [24]. The matter source of sediment from land or ocean and fluctuation of relative sea level on the coast of Weizhou Island have not been well estimated. Some research has been carried out to assess the fluctuation of relative sea level on Weizhou Island [25], and a number of studies have investigated the trophic status of corals and permeable sandy sediment using carbon as an indicator [26,27]. However, a detailed understanding of the matter sources of coastal sediment and their environmental implications for relative sea level fluctuations on Weizhou Island is still largely lacking. In this paper, inorganic carbon and its carbon and oxygen isotope analysis, and organic carbon and nitrogen and their isotope tracing technology, were used together to understand the geochemical characteristics of the coastal area of Weizhou Island since the formation in the Holocene to address this gap in the current understanding. We aim to explore the material provenance of Weizhou Island through the changes in TIC, $\delta^{13}C_{carb}$, $\delta^{18}O_{carb}$, C/N, $\delta^{13}C_{TOC}$, and $\delta^{15}N_{TN}$ to assess the suitability of these proxies as indicators for the inorganic carbon formation process, organic matter source, and environmental evolution process regarding relative sea level fluctuation and dominant vegetation type since the late Holocene. This may provide information about sediment input for the growth and development of Weizhou Island coral reefs and a case of carbon and nitrogen component cycling in coastal zones.

## 2. Materials and Methods

### 2.1. Sample Collection

For the distribution of inorganic carbon content and its isotopes, and the organic carbon and nitrogen contents and their isotope components, in the coastal sediments of Weizhou Island, a total of 57 sediments were collected from 9 typical sedimentary profiles in multiple directions on Weizhou Island on April 25th from 10 a.m. to 4 p.m., 2017. In addition, some of the coral samples of C1 were supplemented on June 28th at the same interval, 2017. Core WZ-35 is located on the laterite platform on the island; A1, A2, B1, C1, and D1 are located in the intertidal zone; and A3, C2, and D2 are located in the supra-tidal zone. Obvious coral clastic interlayers are present in A1, B1, and D1, but C1 contains beach rock. After sampling, the sediment and coral debris were packaged and marked separately, and all samples were bagged, brought back to the laboratory, and stored in a refrigerator at −4 °C to prevent oxidation. The specific sampling profile location, core information, and measurement items are shown in Figure 1 and Table 1.

**Table 1.** Information of core sampling on Weizhou Island.

| Core | Depth (cm) | Number of Sample | Measured Analysis | Location |
|------|-----------|------------------|-------------------|----------|
| A1 | 156 | 14 | TIC, TOC, TN and 14C age | 21°03′55.86″ N, 109°06′20.40″ E |
| A2 | 144 | 7 | TIC, TOC and TN | 21°03′48.07″ N, 109°06′22.31″ E |
| A3 | 70 | 3 | TIC, TOC and TN | 21°03′45.54″ N, 109°06′33.56″ E |
| B1 | 210 | 6 | TIC, TOC, TN and 14C age | 21°03′40.10″ N, 109°08′16.04″ E |
| C1 | 30 | 3 | TIC, TOC, TN and 14C age | 21°01′42.93″ N, 109°07′47.41″ E |
| C2 | 168 | 7 | TIC, TOC and TN | 21°01′42.97″ N, 109°07′36.08″ E |
| D1 | 80 | 6 | TIC, TOC, TN and 14C age | 21°01′46.72″ N, 109°05′11.11″ E |
| D2 | 250 | 6 | TIC, TOC and TN | 21°01′59.91″ N, 109°05′25.48″ E |
| WZ-35 | 600 | 5 | TIC, TOC and TN | 21°02′19.40″ N, 109°05′12.25″ E |

### 2.2. Sample Analysis

The samples were classified and processed before the inorganic carbon and its carbon and oxygen isotopes, organic carbon and nitrogen contents and isotopes, and AMS (Accelerator Mass Spectrometry) radiocarbon dating were analyzed. Each sample was placed in a glass dish before being placed in a freezer for 2 h to pre-freeze the water in the sample as ice; then, the sample was placed in a freeze dryer for vacuum freeze-drying for 24 h, removed, and ground to 200 mesh powder with an agate mortar. However, before TOC, TN, and their isotope ($\delta^{13}C_{TOC}$ and $\delta^{15}N_{TN}$) measurements, it was necessary to

add 6 M hydrochloric acid and allow the sample to soak for 6 h to remove carbonate; then, the sample was washed with deionized water to elute the residual hydrochloric acid until the pH was neutral. Finally, using the freeze-drying method, 25–50 mg powder was dried and weighed for analysis.

The ground sample was analyzed with a Thermo NE1122 elemental analyzer for total carbon (TC) and total nitrogen (TN) contents, and the error was less than 0.2%. The TOC in the acid-treated sample was measured by a Thermo NE1122 elemental analyzer, and the difference between the two was the TIC, that is, $TIC = TC - TOC$ [28]. According to the repeated measurement results of the standard substance by this instrument, the relative error was within 0.2%. The carbon and oxygen isotopes in the carbonate of the sediment samples were titrated with phosphoric acid to produce $CO_2$ gas, and the $\delta^{13}C_{carb}$ and $\delta^{18}O_{carb}$ were measured by a Thermo Delta Plus Advantage isotope mass spectrometer with an accuracy of ±0.2%. The organic carbon and nitrogen isotopes must be accurately measured. $\delta^{13}C_{TOC}$ and $\delta^{15}N_{TN}$ were, respectively, measured by $CO_2$ and $N_2$ after the gas was placed in the Thermo Delta Plus Advantage via the carrier gas (such as high purity He gas), of which the $CO_2$ and $N_2$ were obtained by the powder samples burned in the element analyzer with high temperature before oxidation and purification. The results were calculated according to the V-PDB (Vienna Pee Dee Belemnite) standard ratio [29,30]. Every 12 samples were calibrated with 2 standard samples to examine the measurement error, which was not more than 0.2%. The sample analysis was carried out on the shared test platform of the Third Institute of Oceanography, Ministry of Natural Resources.

The coral debris samples in the sediments were sent to the American Beta Laboratory for AMS radiocarbon dating, and the age of the coral debris in the coastal sediments of Weizhou Island was dated as the age data of the same sample of the coastal sediments of Weizhou Island to explain its chronological significance.

## 3. Results

### 3.1. Inorganic Proxies of Costal Sediments on Weizhou Island

#### 3.1.1. Distribution of TIC in Coastal Sediments

The results of inorganic carbon in the coastal sediments of Weizhou Island show that the average value of TIC is between 0.02% and 10.52% (mass percentage, Figure 2), and polarized between intertidal and supra-tidal. Compared to the schematic diagram of the coastal sediment sampling station of Weizhou Island in Figure 1, the TIC contents of all samples in profiles A1, A2, B1, C1, and D1 are above 3% and between 5.09% and 7.33%, on average. Additionally, the TIC content of sample A1-8 (depth of 84 cm) is 10.52%, which is close to the 12% value of the calcium carbonate inorganic component. It is significantly higher than that of the supra-tidal profile (A3, C2, and D2), whose average TIC content is generally close to zero, due to the cover of effect of tides and waves, but there is a small amount in the lower profile only in C2. The TIC mostly changes periodically from the surface to the bottom in the intertidal zone.

#### 3.1.2. The Distribution of $\delta^{13}C_{carb}$ and $\delta^{18}O_{carb}$ in the Coastal Sediments

The stable carbon isotopes of carbonates range from −12.67‰ to −0.01‰ (Figure 3), and the distribution corresponds to the TIC. The sediment $\delta^{13}C_{carb}$ in the intertidal zone is relatively positively biased, with most values higher than −3‰, whereas the supra-tidal zone is negatively biased, suggesting that δ13Ccarb shows obvious differences from the coast to offshore areas, such as TIC. However, anomalies appear in the low parts of A3 and C2 compared with those cores in the supra-tidal zone. These indicate the formation of carbonate [10].

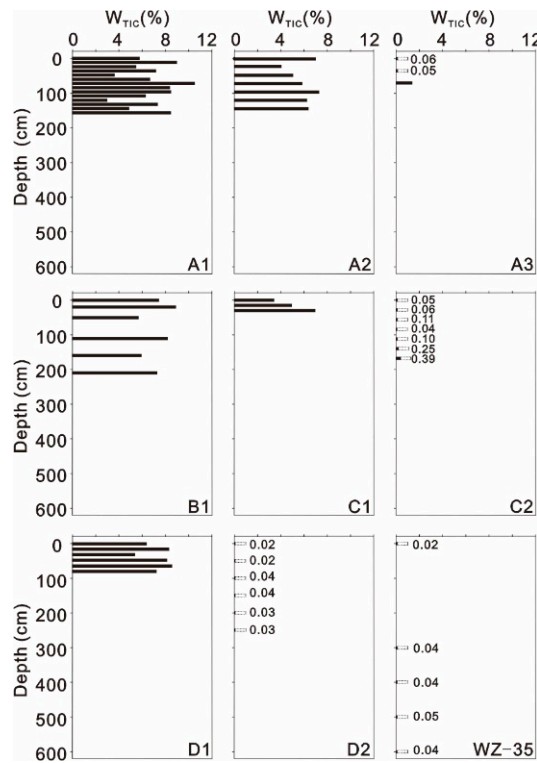

**Figure 2.** Distribution of TIC of the core sediments

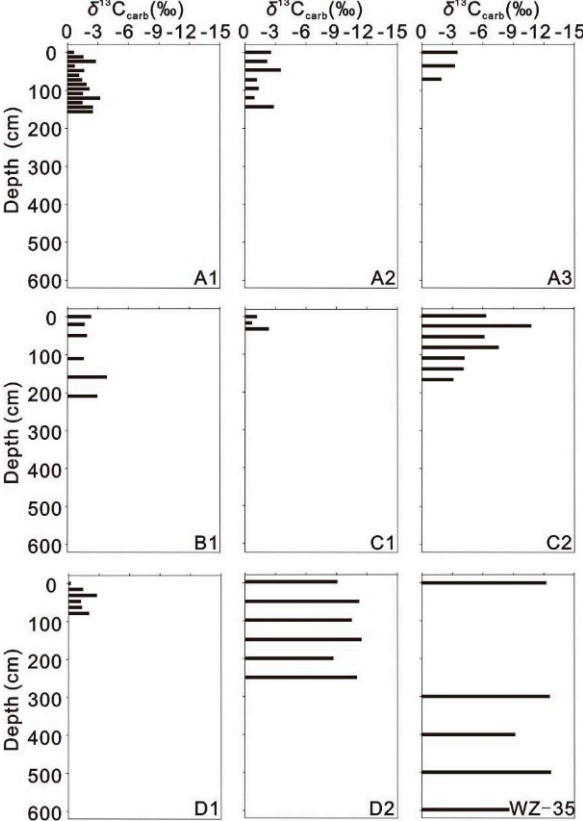

**Figure 3.** Distribution of δ13Ccarb of the core sediments.

The carbonate oxygen stable isotopes range from −20.20‰ to −3.49‰ (Figure 4), and the distribution shows a positive correlation with $\delta^{13}C_{carb}$ ($R^2$ = 0.7478). The intertidal profile $\delta^{18}O_{carb}$ is positively biased, at approximately −5‰, and the continental profile has a significant negative bias. However, the negative bias is at low parts of A3 and C2 compared with the supra-tidal cores for the reason stated above for $\delta13Ccarb$.

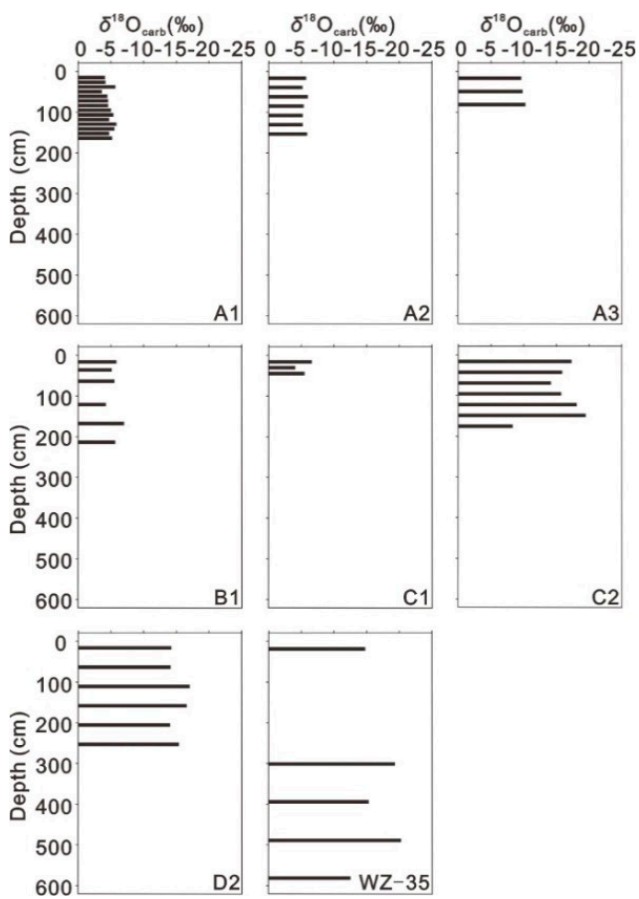

**Figure 4.** Distribution of $\delta18O_{carb}$ of the core sediments.

*3.2. Organic Proxies of Coastal Sediments on Weizhou Island*

3.2.1. Distributions of TOC and TN in the Coastal Sediments

The TOC values of all samples do not exceed 2% and 0.09–1.25% (Figure 5), on average, and the trend fluctuates and increases from the surface to the bottom, except at B1 and D1, which show the opposite trend. The TOC content of cores is 0.47% higher on average on the northern coastline than on the southern coastline, which is especially prominent in the intertidal region as a result of the difference in deposition [14].

The results of organic nitrogen content in the sediments of Weizhou Island show that the TN contents of sediments in all profiles are between 0.02% and 0.17% (Figure 6), and the TN distribution has a relatively consistent relationship with that of TOC, indicating that they may not be subject to human interference.

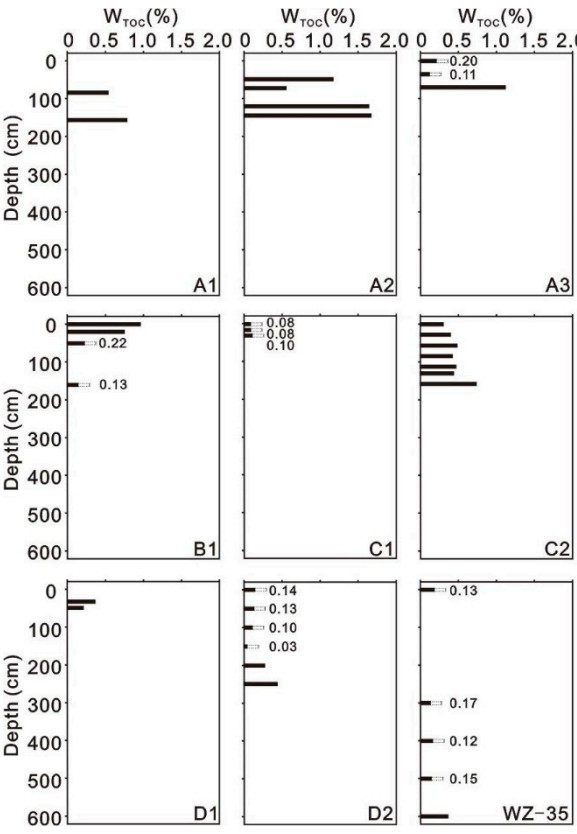

**Figure 5.** Distribution of TOC of the core sediments.

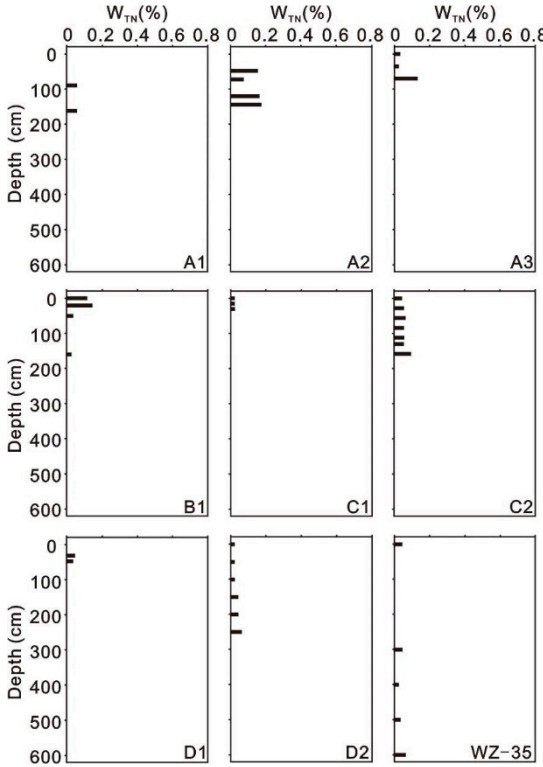

**Figure 6.** Distribution of TN of the core sediments.

### 3.2.2. Sediment $\delta^{13}C_{TOC}$ and $\delta^{15}N_{TN}$ Distributions

$\delta^{13}C_{TOC}$ ranges from −14.96‰ to −27.26‰ (Figure 7). In the intertidal zone, $\delta^{13}C_{TOC}$ is positively biased in A1, A2, B1, C1, and D1, and anomalies appear only in the low part of A2 as a result of the input of $C_3$ plants. In the supra-tidal zone, the A3, C2, and D2 $\delta^{13}C_{TOC}$ distributions are relatively concentrated. Only in the low part of D2 is there $\delta^{13}C_{TOC}$ enrichment due to marine organic matter input. Overall, not all δ13CTOC is enriched with a high content of TOC, which is controlled by the organic matter source, including $C_3$ and $C_4$ plants and marine algae [11].

Compared with the laterite platform of profile WZ-35, the coastal sediments of Weizhou Island have significant differences in TIC, $\delta^{13}C_{carb}$, $\delta^{18}O_{carb}$, TOC, TN, $\delta^{13}C_{TOC}$, and $\delta^{15}N_{TN}$ because of the difference in formation, indicating that the coastal sediment is not basalt weathered soil generated in situ but rather a secondary mixture.

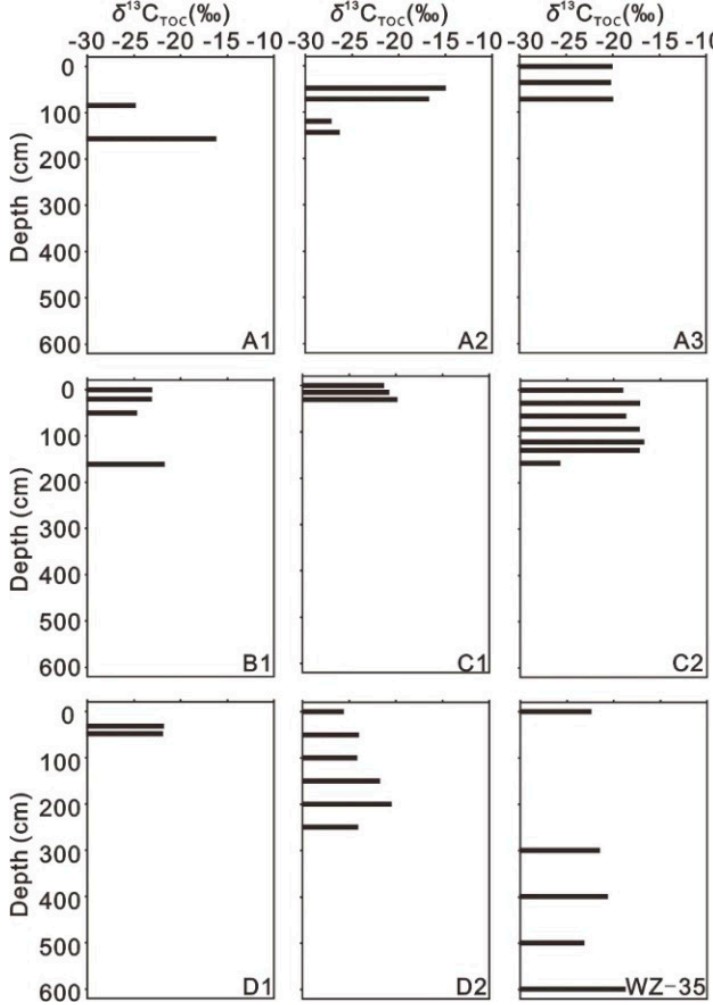

**Figure 7.** Distribution of δ13CTOC of the core sediments.

$\delta^{15}N_{TN}$ ranges from −14.38‰ to 4.12‰ (Figure 8), with an average value of −1.91‰, which is relatively correlated with $\delta^{13}C_{TOC}$. However, there are exceptions at the low parts of A1, A2, and C2 for the changes of organic matter source. Overall, the distribution of $\delta^{15}N_{TN}$ in the intertidal profile is low, but is enriched in the supra-tidal zone.

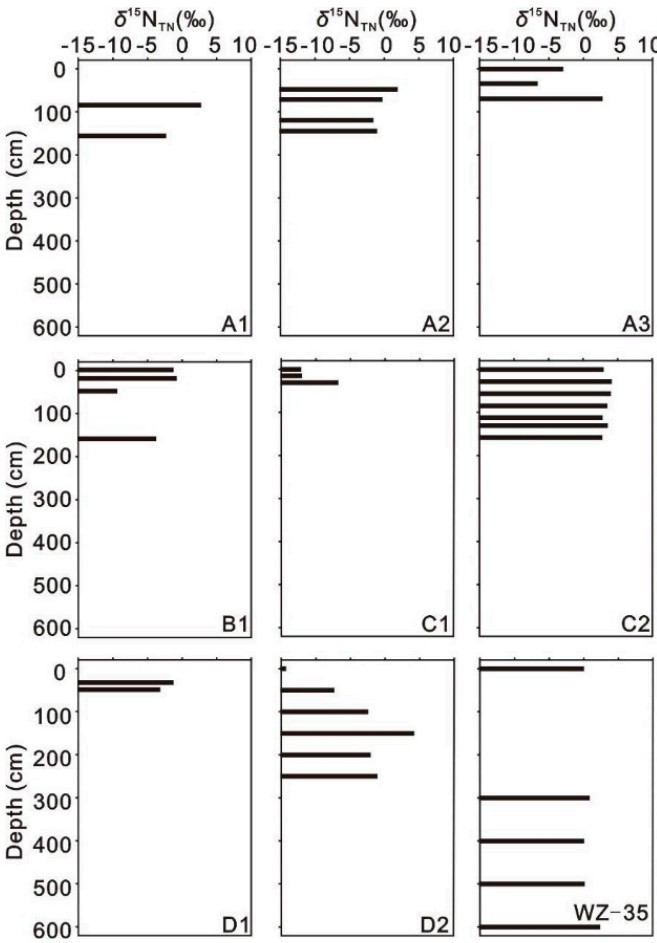

**Figure 8.** Distribution of δ15NTN of the core sediments.

### 3.3. AMS Radiocarbon Dating Analysis

The coral clastic samples of coastal sediments on Weizhou Island ranged from 450 to 2750 cal. a BP (Table 2), with an average age of 1392 ± 30 cal. a BP. They belong to the late Holocene (Table 2) $Q_4^3$ sediment. The stratum was deposited in order from the bottom to the surface, and the geological age increased with depth. The average deposition rate of C1 is approximately 0.13 mm/a, indicating that there is a relatively stable deposition environment. However, the geological age of the sediments in the D1 profile is reversed.

**Table 2.** Radiocarbon dating data of coral debris sample of the core sediments.

| Sample Code | Depth (cm) | Sample Number | AGE(BP) (Years) | Material Measured |
|:---:|:---:|:---:|:---:|:---:|
| A1-1 | 0 | 470544 | 720 ± 30 | coral |
| A1-8 | 84 | 470541 | 1000 ± 30 | coral |
| B1-2 | 20 | 470547 | 1290 ± 30 | coral |
| B1-5 | 160 | 470545 | 2310 ± 30 | coral |
| C1-1 | 0 | 470549 | 450 ± 30 | coral |
| C1-2 | 15 | 470538 | 1490 ± 30 | coral |
| C1-3 | 30 | 470537 | 2750 ± 30 | coral |
| D1-1 | 0 | 470552 | 1220 ± 30 | coral |
| D1-4 | 48 | 470551 | 1200 ± 30 | coral |
| D1-5 | 64 | 470550 | 1050 ± 30 | coral |

## 4. Discussion

*4.1. Environmental Significance of Inorganic Proxies under Evaporation*

4.1.1. Source analysis of TIC content

Inorganic carbon is an important part of coastal sediments, and has always played an important role in tracing the source, deposition, circulation, and decomposition of coastal materials (including biomass and sandy coast). The main source of inorganic carbon on a sandy coast is the input of particulate or dissolved substances carried by the tide; the second is the residual deposits of the dead bodies of beach organisms; the third is the input of marine steam transport and evaporation in the pores of sandy beaches; and the fourth is the input of terrestrial debris carried by surface runoff [8,31].

The coastal sediment TIC of Weizhou Island increases rapidly from the coast to offshore areas (Figure 2), and the TIC content in the supra-tidal zone is close to zero, indicating that marine contributions dominate the carbonate input of coastal sediments. In the ocean-influenced dominant intertidal zone, the carbonates carried by marine spray and seawater are transported to the sandy coast under the transportation of waves and remain in the pores of the beach. The high-salinity pore water is supersaturated by high-temperature evaporation, forming a microwater rock environment. The chemical equation is as follows:

$$Ca^{2+} + 2HCO_3^- \Leftrightarrow CaCO_3 + CO_2 + H_2O$$

Evaporation increases the carbonate ion concentration and shifts the balance to the right, which is conducive to the precipitation of carbonate and preservation resulting in enrichment in the sediment. In addition, in the high-abundance carbonate sedimentary profile dominated by the marine contribution, the surficial sediment TIC of C1 is significantly lower than that in other high-abundance carbonate profiles because the fresh water carried by the Shipan River dilutes the beach pore water, which contributes to a lower salinity and suppresses this progress. The distribution of a small amount of carbonate at the low part of C2 may be due to the intrusion of seawater following the increase in carbonate ions in the pore water to precipitate carbonate. Weizhou Island is an island of volcanic origin covered with basalt-weathered laterite platform deposits and Holocene residues. WZ-35, composed of laterite, has nearly no TIC deposits, indicating that the island is not the source of soil inorganic carbon. In addition, the abundant surface runoff has a strong desalination and erosion effect on the surface sediments on Weizhou Island, so the TIC in the supra-tidal zone with a limited marine contribution is close to zero. The carbonate content in sandy beaches can determine the degree and range of marine contributions.

4.1.2. Ocean Influence Dominated the Area of Distributions of $\delta^{13}C_{carb}$ and $\delta^{18}O_{carb}$

In a sedimentary environment dominated by marine contributions, the deposition process of inorganic carbon is closely related to the evaporation and concentration processes of water. Among isotopic kinetic fractionations, lighter isotopes are usually preferentially enriched in the reaction products before the reaction reaches equilibrium. Therefore, $^{12}C$ is preferentially enriched in the reaction products in evaporation or plant photosynthesis. The general rule is that heavy isotopes move into the most tightly bonded phase. Carbonate tends to be rich in $^{18}O$ because O prefers to be bound to small and high-priced $C^{4+}$ ions by ionic bonds [10].

The coastal sediments of Weizhou Island show obvious differences in the distributions of $\delta^{13}C_{carb}$ and $\delta^{18}O_{carb}$ (see in Section 3.1.2). The intertidal zone carries sufficient seawater evaporation and concentration from tides and waves following a faster diffusion rate of lighter $^{12}C$ and $^{16}O$. This process leads to the abundant accumulation of $^{13}C_{carb}$ and $^{18}O_{carb}$ in carbonate. In the supra-tidal zone, the limited marine contribution affected the lack of sufficient seawater evaporation and concentration, and the absorption of inorganic carbon by terrestrial plants, in addition to the input of rainwater and

surface runoff, support abundant $CO_2$, which enriches $^{12}C$ and $^{16}O$ from air, leading to a decrease in the components of $\delta^{13}C_{carb}$ and $\delta^{18}O_{carb}$.

*4.2. Organic Matter Sources of Coastal Sediment on Weizhou Island*

4.2.1. Source Analysis of Organic Carbon and Nitrogen

According to the geochemical signal response for different sources, C/N ratios are often used to distinguish the sources of organic matter [32]. When the C/N ratio is greater than 12, it indicates that the organic matter comes from terrestrial contributions [33]. The C/N ratios of typical marine matter are less than 8 [34], whereas those between 8–12 are a mixed source of materials from land and sea [18,32]. The C/N ratios in the coastal sediments of Weizhou Island are in the range 0.90–14.42 (Table 3), with an average value of 7.02. Most of the samples have been imported by marine organic matter.

**Table 3.** List of C/N ratios of the core sediments.

| Sample Code | Depth (cm) | C/N | Sample Code | Depth (cm) | C/N | Sample Code | Depth (cm) | C/N |
|---|---|---|---|---|---|---|---|---|
| A1-1 | 84 | 11.38 | B1-5 | 160 | 6.32 | D1-4 | 48 | 7.45 |
| A1-7 | 156 | 14.42 | C1-1 | 0 | 4.21 | D2-1 | 0 | 7.55 |
| A2-1 | 48 | 7.55 | C1-2 | 15 | 3.92 | D2-2 | 50 | 5.92 |
| A2-2 | 72 | 7.96 | C1-3 | 30 | 4.59 | D2-3 | 100 | 4.87 |
| A2-4 | 120 | 9.94 | C2-1 | 0 | 8.21 | D2-4 | 150 | 0.90 |
| A2-5 | 144 | 9.55 | C2-2 | 28 | 7.21 | D2-5 | 200 | 6.48 |
| A3-1 | 0 | 6.92 | C2-3 | 56 | 7.32 | D2-6 | 250 | 7.19 |
| A3-2 | 35 | 6.67 | C2-4 | 84 | 7.58 | WZ35-1 | 0 | 3.50 |
| A3-3 | 70 | 8.22 | C2-5 | 112 | 8.46 | WZ35-2 | 300 | 4.94 |
| B1-1 | 0 | 8.73 | C2-6 | 140 | 8.73 | WZ35-3 | 400 | 5.08 |
| B1-2 | 20 | 5.47 | C2-7 | 168 | 8.28 | WZ35-4 | 500 | 5.03 |
| B1-3 | 50 | 6.94 | D1-3 | 32 | 8.77 | WZ35-5 | 600 | 6.31 |

4.2.2. Correlation of $\delta^{13}C_{TOC}$ and $\delta^{15}N_{TN}$ and Their Environmental Significance

Because the process of soil degradation may have a significant impact on C/N and $\delta^{13}C_{TOC}$, choosing a single factor to consider is likely to result in an incorrect understanding of the environmental evolution process [21]. Therefore, the C/N attached to the $\delta^{13}C_{TOC}$ relationship diagram is a more effective way to qualitatively analyze the source of organic matter [19,32,35,36]. Generally, when different organisms carry out life activities, the application of carbon is different due to inner or outer conditions, such as enzymes and the existing environment, resulting in the differentiation of organic carbon isotopes of synthetic products [37]. $C_3$ plants preferentially absorb the $^{12}C$ component during photosynthesis, resulting in plant organic carbon $\delta^{13}C_{TOC}$ values lower than the $\delta^{13}C_{TOC}$ of $CO_2$ in the air (−8‰), ranging from −26‰ to −28‰ [38]. Planktonic algae $\delta^{13}C_{TOC}$ is positively biased, between −19‰ and −22‰ [39]. $C_4$ plants have vascular bundle sheaths and the phosphoenolpyruvate enzyme (participating in photosynthesis) acts on the absorption of $CO_2$, which is different from the former. Thus, their $\delta^{13}C_{TOC}$ range is approximately from −17‰ to −9‰ [11,37,40].

The organic carbon content of sediments gradually decreases with increasing profile depth, which is consistent with the changes in sediment nitrogen content [37]. In this paper, TN and TOC were significantly positively correlated ($R^2$ = 0.9104) in the scatter plots, suggesting that the two had a common source, but Figure 9 shows that the TN axis had an intercept of b = 0.0110, not b = 0, indicating that TN contained inorganic nitrogen, which may decrease the C/N. Combining the C/N and $\delta^{13}C_{TOC}$ to distinguish the source of organic matter (Figure 10) in the intertidal zone, terrestrial contributions of organic matter occurred in samples A1-7 (84 cm) and A1-14 (156 cm). The C/N of A2 decreased from the bottom to the top, and $\delta^{13}C_{TOC}$ became lighter and approached that of terrestrial $C_3$ plants at low levels. However, the middle $\delta^{13}C_{TOC}$ became heavier at the middle of A2, indicating that the composition of $C_3$ plants from organic matter sources in A2 was reduced. Furthermore, according to the contribution

variation and relatively consistent sedimentary environment of A1 and A2, we speculate that terrestrial contributions of organic matter occurred below the middle of A1. B1 was located in the transition zone between phytoplankton and $C_3$ plants, approaching the marine plot, indicating that the overall organic matter was mainly from marine contributions, but it was affected by the input of terrestrial $C_3$ plant detritus. C1 was completely within the range of marine algae. C/N and $\delta^{13}C_{TOC}$ in D1 approached the marine phytoplankton unit. In the supra-tidal zone, the C/N of most samples was low but significantly deviated from the marine phytoplankton interval. Combined with the human geographical history of Weizhou Island, it is speculated that in addition to the mix of inorganic nitrogen, it is likely that the C/N is reduced due to the sowing of nitrogen fertilizer on agricultural land. This result can also be seen in the fluctuating correspondence between $\delta^{15}N_{TN}$ and $\delta^{13}C_{TOC}$ in the supra-tidal zone. The C/N and $\delta^{13}C_{TOC}$ of A3 fall within the range of marine phytoplankton as a whole. All C2 samples are in the transition zone between marine and terrestrial sources. C2-7 indicates the transition zone between marine algae and $C_3$ plants, whereas the remaining points are in the transition zone between marine algae and $C_4$ plants. It is likely that the replacement of terrestrial organic matter sources between C2-7 and C2-6 represents a succession of plant communities. C/N and $\delta^{13}C_{TOC}$ in D2 are located in the transitional range of sea sources and $C_3$ plants. Combined with the whole profile, the input of organic matter from the marine source in the lower part is inferred from storm surge deposition.

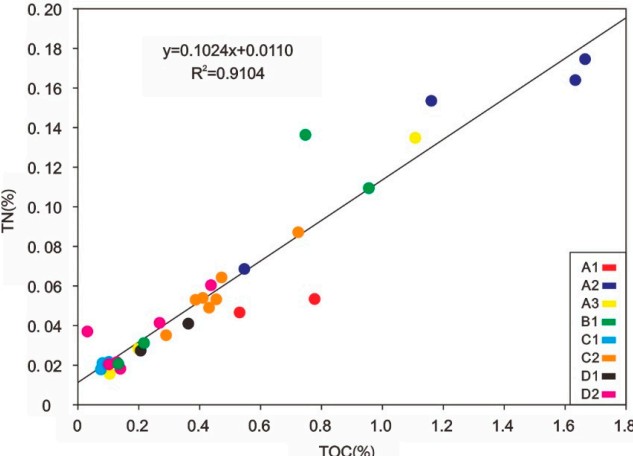

**Figure 9.** The relationship between total organic carbon (TOC) and total nitrogen (TN) of the core sediments on Weizhou Island. The color that represents the core is shown in the lower right corner of the figure.

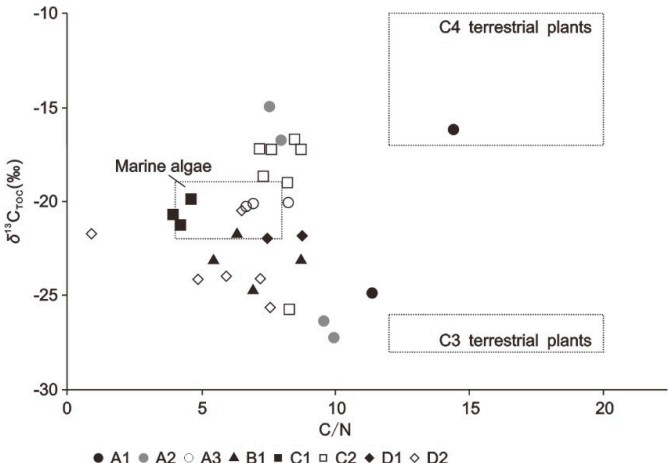

**Figure 10.** Distribution of δ13CTOC and C/N of the core sediments in the correlation plots of different organic sources [11,38,39].



In addition, $\delta^{15}N_{TN}$ can indicate the source of organic matter. Phytoplankton $\delta^{15}N_{TN}$ ranges from 4‰ to 10‰, fluvial and terrestrial organic matter $\delta^{15}N_{TN}$ varies greatly, but most are lower than that of marine organic matter, which ranges from −10‰ to 10‰. Additionally, $\delta^{15}N_{TN}$ from human pollutants is usually positive [36,41]. In this study, the overall growth trend between $\delta^{15}N_{TN}$ and $\delta^{13}C_{TOC}$ (Figure 11), with the exception of individual points that may be interfered with by man-made pollution sources, is roughly the same, indicating that the two are likely to come from the same provenance. In the intertidal zone, $\delta^{15}N_{TN}$ is negatively biased, which confirms the existence of terrestrial organic matter [42,43].

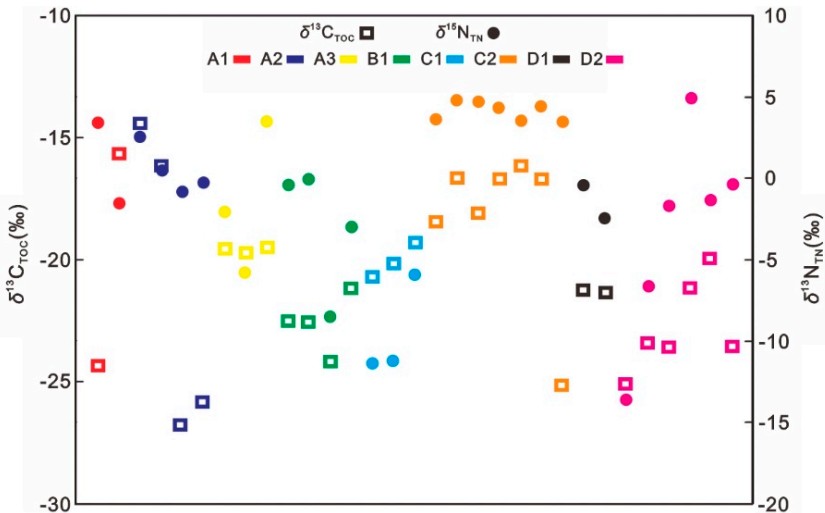

**Figure 11.** Distribution of δ13CTOC and δ15NTN of core sediments.

### 4.3. The Regression of the Coastal Environment on Weizhou Island

The coral clastic samples of coastal sediments on Weizhou Island are all Holocene sediments, and the deposition rates of A1 and B1 are relatively high (Figure 12). Because A1 and B1 are located in the northwest and northeast wave shadows and the low-lying area of the island, respectively, coupled with the unloading force of the developed coral reef on waves, coastal erosion is weak, and sediments are transported and collected by coastal currents and surface runoff. C1 is located in the southeastern corner of the island, which is affected by the monsoon. Waves cause strong coastal erosion in autumn and summer, and the sedimentation effect is significantly lower than the former.

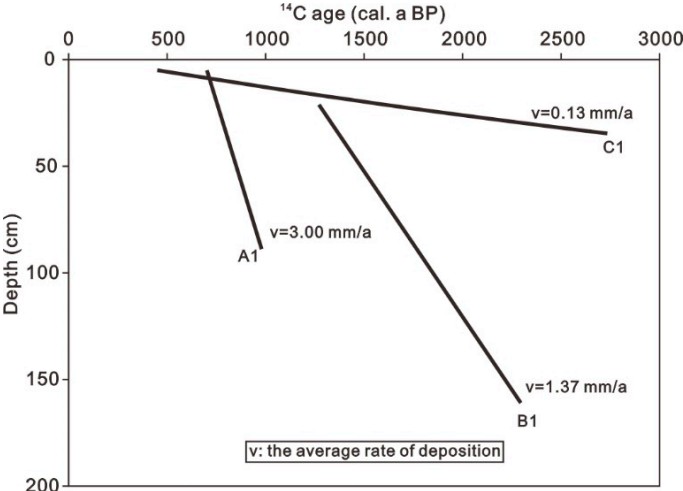

**Figure 12.** $^{14}C$ dating age of the coral sediments with their average deposition rate.

The TIC, $\delta^{13}C_{carb}$, $\delta^{18}O_{carb}$, C/N, $\delta^{13}C_{TOC}$, and $\delta^{15}N_{TN}$ values of coastal sediments on Weizhou Island and the analysis of the sedimentary characteristics of the island's landform showed that it obviously obtained a supply of sediment from the ocean (Figure 13). The intertidal TIC was positively correlated with $\delta^{13}C_{carb}$ and $\delta^{18}O_{carb}$ (see Section 3.1), indicating that the ocean played an important role in the intertidal zone. However, the increase in the input of terrestrial organic matter at low parts of A1 and A2, and the input of organic matter from the marine source in A3 (Figure 10), reflected the obvious regression trend in northern Weizhou Island. The sediments of B1 were mainly imported from marine sources, and a small amount of $C_3$ plant debris was mixed in the organic matter. Therefore, the northern coast has advanced approximately 500 m offshore, which proves that the northern coast of Weizhou Island is accumulating and that the coastline is expanding [23]. However, this could be accompanied by a reduction of $C_3$ plant input according to the organic matter source of the upper parts of A2, A3, and B1. Xiong et al. (2018) [44] reported that the sea level in the northern part of the South China Sea had not changed significantly during the past 7000 years, and Yao et al. (2009) [45] suggested that the Middle Holocene had been affected by tectonic movement and sedimentation. Although the sea level has not changed significantly in the South China Sea, it is mainly regressive, and the land area in the northwest has been expanding. It can be inferred that in the surroundings of Weizhou Island, the relative sea level has been declining since the late Holocene due to tectonic rise. The estuary area of the C1–C2 section was strongly affected by the waves of the open sea, but due to the influence of the river, the TIC was rapidly reduced due to the desalination of the sea. $^{13}C_{carb}$ and $\delta^{18}O_{carb}$ were depleted (see Section 3.1.2), and the input of organic matter changed from ocean influence dominated to land–sea mixed sources because the ocean's role in the estuary area was sharply weakened. Following the sediment input from the D1–D2 section, enhanced by the reduction of marine matter input to the shore and the surface runoff from abundant rainfall (Figure 10), the terrestrial contribution of D2 was enhanced, but the input signal of terrestrial organic matter at the low part of D2 was weakened, which, in addition to the reversal age of D1, reflects the disturbance caused by storm surges to coastal sediments. These results may provide information about the material sources of coastal zones and sedimentary environments for the growth and development of coral reefs.

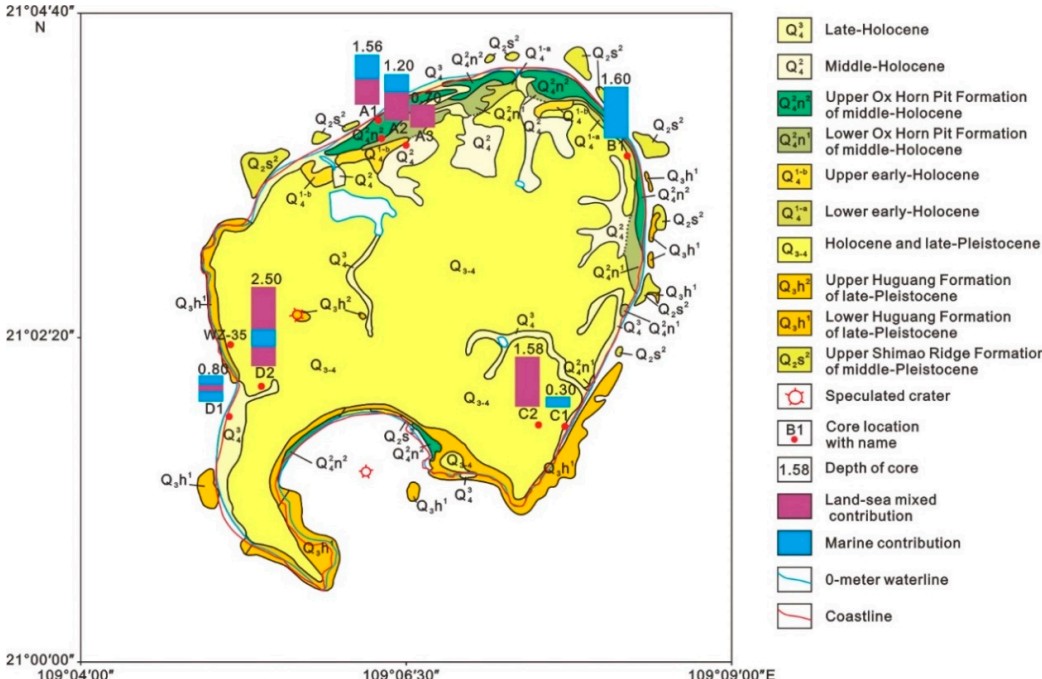

**Figure 13.** Schematic diagram of the source of the core sediments on Weizhou Island. Different material sources in cores are shown in different colors. The stratum information is also color-coded in the figure. In addition, the coastline and 0-m waterline are marked.

The C1 coral clastic profile is located on the beach rock of the coastal beach. The beach rock is the petrified sediment of the ancient intertidal zone or the wave splash zone, and its formation represents the location of the ancient sea level [46,47]. Taking C1-3 as the ancient sea level point and referring to the radiocarbon dating age (Table 2), the average sea level decline rate of Weizhou Island has been approximately 2.07 mm/a in the past 2750 years, which is similar to the value of 1.6–2.1 mm/a estimated by Mo Yongjie [48]. This finding is consistent with the Holocene sea level decline in the South China Sea [49].

## 5. Conclusions

The carbon and nitrogen contents, and their related isotope measurements, of coastal sediments of Weizhou Island showed that the TIC content was between 0.02% and 10.52%, and the ranges of $\delta^{13}C_{carb}$ and $\delta^{18}O_{carb}$ were −12.67‰ to −0.01‰ and −20.20‰ to −3.49‰, respectively. TIC, $\delta^{13}C_{carb}$, and $\delta^{18}O_{carb}$ were enriched in the intertidal zone. C/N was between 0.90 and 14.42, and the ranges of $\delta^{13}C_{TOC}$ and $\delta^{15}N_{TN}$ were −27.26‰ to −14.96‰ and −14.38‰ to 4.12‰, respectively. The burial of organic matter and its isotopes was different in different profiles, and the isotope fractionation effect was remarkable.

According to the identification of the geochemical information of coastal sediments on Weizhou Island, the main source of TIC was the concentration of water in the intertidal zone, which was dominated by ocean actions such as tides and waves. The dominant vegetation type on the coast of Weizhou Island is a mixture of $C_3$ and $C_4$ plants. The decline in terrestrial organic matter in northern profiles A1, A2, and B1 proved that the important source of the "northern deposition" of Weizhou Island came from the island land, which may have followed the reduction of $C_3$ plant input. The decrease in TIC and the change in organic matter in the C1–C2 section reflected the controlling effect of rivers in the estuary area. The enhanced marine source signal at the low part of D2 was speculated to be evidence of residual organic matter in the marine source swept by the storm surge.

The coast of Weizhou Island is generally regressing, and the source is mainly from the ocean. The northern coast is an accumulative landform with an enhanced terrestrial contribution, but there are weak deposits on the southeastern coast. Based on the dating data of the C1 profile, it is estimated that the average rate of sea level decline in the past 2750 years has been approximately 2.07 mm/a. It is possible to distinguish sources of inorganic and organic material on an island distant from other land and, hence, make an assessment of the evolution of the sedimentary environment.

**Author Contributions:** Designed the study, wrote the main manuscript and prepared all figures: Z.T.; contributed to the improvement of the manuscript: C.C.; processed the data: H.Q. and J.Y.; collected the data: K.T. and Y.S.; all authors reviewed the manuscript. All authors have read and agreed to the published version of the manuscript.

**Funding:** This research is funded by the National Natural Science Foundation of China (Grant No. 42076058, 41930538 and 41406059), the Scientific Research Foundation of Third Institute of Oceanography, MNR (Grant No. 2019006), and Special Funds for Scientific Research on Marine Public Causes (Grant No. 201505012-5).

**Acknowledgments:** The authors would like to express their sincere thanks to all those who have offered support.

**Conflicts of Interest:** The authors declare no conflict of interest.

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
