# Peer review of "Distribution of Carbon and Nitrogen as Indictors of Environmental Significance in Coastal Sediments of Weizhou Island, Beibu Gulf"

_water, doi:10.3390/w12113285_

Round 1

Reviewer 1 Report

Dear Authors,

I like your paper! Unfortunately my experience with tropical type of sediments in insufficient for more detailed comments but I am able to recognize well provide technical, methodological and interpretational level of your proposed paper. 

My comments is related mainly on some suggestions:

Row 146 - Try to explain why anomalies at low parts of A3 and C2?

Row 152 - As well as above!

Rows 159, 171, 177,189 - As well as above! Why?

Fig 9. Mark each particular samples as well as on Fig. 10 or 11. 

In introduction as well as in conclusion you can pointed out why results of your paper is important for study islands for example ecological point of view, climate changes (influence of storms, tidal events) etc. suggest your opinion of future scientific projects related to your and similar islands, compare your results with other nearby islands or something else.

Best regards....  

Author Response

Response to Review Comments (water-994928)

We are greatly encouraged by the positive comments by the Guest Editor and the Chief Editor in view of our previously submitted manuscript through a peer review. A special thanks to the Guest Editor, the Chief Editor, and three anonymous reviewers for their deep and thorough reviews. We have tried our best to modify the previous submission of this manuscript regarding all the detailed comments and constructive suggestions.

To better show how and where the earlier version is thoroughly revised, both the “track changes” and clean version of the manuscript based on the previous submission are provided. All line numbers mentioned in this detailed reply belong to the manuscript with “track changes”.

Reviewer #1:

I like your paper! Unfortunately my experience with tropical type of sediments in insufficient for more detailed comments but I am able to recognize well provide technical, methodological and interpretational level of your proposed paper.

My comments is related mainly on some suggestions:

Row 146 - Try to explain why anomalies at low parts of A3 and C2?

Row 152 - As well as above!

Rows 159, 171, 177,189 - As well as above! Why?

Response: These suggestions were greatly appreciated. We have taken these suggestions and added more explanation for the distribution characteristics of these geochemical elements. See the detail revision in lines 174-175, 186, 194, 202-204, 210-211, 216-223, 229 and 236 in the revised manuscript with track changes, respectively.

Fig 9. Mark each particular sample as well as on Fig. 10 or 11.

Response: Thanks to this suggestion, and it has been modified by colored circle in Figure 9.

In introduction as well as in conclusion you can pointed out why results of your paper is important for study islands for example ecological point of view, climate changes (influence of storms, tidal events) etc. suggest your opinion of future scientific projects related to your and similar islands, compare your results with other nearby islands or something else.

Response: We are truly appreciated for your constructive and detailed comments. Weizhou Island is a natural testing place, which reserves information about environment evolution without large-scale human disturbance. Accounting to what we have done before, we have concluded more about environment evolution and deposition following the organic matter change. In addition, the highlights we have modified to emphasized the novelty to the community of research. See the detail revision in Highlights and Conclusions of the new manuscript with track changes, respectively.

Reviewer 2 Report

The manuscript is presented correctly, but it should emphasize the novelty of it and the contribution to the scientific community. In addition, the sample collection section should be better explained indicating in which moments the samples have been taken, since the study can only be centered for a certain moment without taking into account the tidal or ocean dependence.

Author Response

Response to Review Comments (water-994928)

We are greatly encouraged by the positive comments by the Guest Editor and the Chief Editor in view of our previously submitted manuscript through a peer review. A special thanks to the Guest Editor, the Chief Editor, and three anonymous reviewers for their deep and thorough reviews. We have tried our best to modify the previous submission of this manuscript regarding all the detailed comments and constructive suggestions.

To better show how and where the earlier version is thoroughly revised, both the “track changes” and clean version of the manuscript based on the previous submission are provided. All line numbers mentioned in this detailed reply belong to the manuscript with “track changes”.

Reviewer #2:

Due to its geochemical approach, this article is interesting and allows to understand the different terrestrial and oceanic interactions leading to coastal sediments of the Weizhou Island. In particular, the authors use the isotopic geochemistry of carbon, nitrogen, and oxygen to trace the origin of the various organic and inorganic inputs. However, the density of information means that more explanations are needed.

Response: We have accepted these comments. To make it more scientific, we have added more contents about the contribution of material source in northern coast and the dominant vegetation type on Weizhou Island. See the detail revision in lines 403-409 and 453-457 of new manuscript with track changes.

Major revision:

Lines 149-150: the distribution shows a positive correlation with δ13Ccarb. Difficult to interpret, a Figure/correlation coefficient are missing

Response: We are sorry for the mistake/confusion in this manuscript and inconvenience it caused in your reading. We would like to indicate that TIC and have a certain correlation with δ13Ccarb and δ18Ocarb in the figures of distribution by comparing the distribution diagram. To make it clear, we have added more comment. In addition, the specific analysis is explained in the following section, 4.1.2 Ocean influence dominated the area of distributions of δ13Ccarb and δ18Ocarb, including why they have this relationship and what kind of process have arose. See the detail revision in lines 186, 191 and 194 in new manuscript with track changes.

Line 164: the TN distribution is nearly consistent with that of TOC. Not obvious, needs more explanation.

Response: We have accepted this suggestion. The organic carbon content of sediments gradually decreases with increasing profile depth, which is consistent with the changes in sediment nitrogen content. In this section, we have talked about the distributions of TOC and TN with explanation in lines 202-204 and 210-211 of the revised manuscript with track changes. The detailed explanation is indicated in the section 4.2 Organic matter sources of coastal sediment on Weizhou Island.

Line 175: which is roughly positively correlated with δ13CTOC. More explanation is required (what is the correlation coefficient?)

Response: We have taken this suggestion seriously. We would like to show that the δ13CTOC is relatively consisted with δ15NTN in this section. This is for the reason that they come from the same source in parts of samples, such as C3 or C4 plant, marine organic matter and land-sea mixed contribution. To make it clear, we have modified with more explanation in lines 216-223 and 229 of new manuscript with track changes.

Minor revision:

In the figure 1 you should emphasize the index 1 corresponds to intertidal higher indexes to supra-tidal

Response: We have accepted this suggestion. We have modified these sentences to make the discussion clear. We have modified it in the Figure 1 of the new manuscript with track changes.

Line 13: AMS radiocarbon dating ages (Accelerator Mass Spectroscopy?)

Response: Yeah, it is Accelerator Mass Spectroscopy.

Line 73: sea level decline (explain)

Response: This is a nice suggestion that has pointed out what we have ignored before. In this paragraph, we would like to briefly introduce Weizhou Island. But sea level decline maybe too ambiguous to introduce, so we replace it by “tectonic uplifting” in line 86 of the new manuscript with track changes.

Line 113: AD isotope mass spectrometer (explain AD)

Response: We have adopted this worth comment and the full name is Thermo Delta Plus Advantage (AD) isotope mass spectrometer. We have made it clear in lines 149 and 152 of the new manuscript with track changes.

Line 135: sample A1-8 Specify the depth

Response: This comment has been taken and we have added the depth of sample in line 172 of the new manuscript with track changes.

Line 143: the distribution is proportional to the TIC. What does it mean?

Response: We have accepted this comment. From the figure, we can see that the distribution of TIC is consistent with δ13Ccarb, which TIC increases with the rich of δ13Ccarb. To make it clear, we have modified it in line 182 of the revised manuscript with track changes.

Lines 145, 152: anomalies appear at the low parts of A3 and C2 I see nothing

Response: We are sorry for the mistake/confusion in this manuscript and inconvenience it caused in your reading. We can see that the distribution of δ13Ccarb and δ18Ocarb of A3 and the low parts of C2 was abnormal in the supra-tidal area. To make it clear, we have modified the sentence to emphasize it in lines 186 and 194 of the revised manuscript with track changes.

Why δ13C Is represented on a decreasing scale in Fig. 3 and on an increasing scale in Fig. 7?

Response: The reason is that some values of δ15NTN are more than 0. We would like to make these distribution figures on a decreasing/increasing scale, but the abscissa in Figure 3 (distribution of δ13Ccarb of the core sediments) and Figure 4 (distribution of δ18Ocarb of the core sediments) can start from zero, which we accustomed to it, while Figure 8 (distribution of δ15NTN of the core sediments) cannot start from zero. We have divided TIC and TOC/TN into two sections, so Figure 3 and Figure 4 is in decreasing scale, but Figure 7 and Figure 8 is opposite to the isotope distributions.

Line 186: The stratum was deposited in order from the surface to bottom? It is the contrary!

Response: We are sorry for the mistake/confusion in this manuscript and inconvenience it caused in your reading. We have modified it to avoid mistake/confusion in line 242 of the revised manuscript with track changes.

Line 228: O prefers to be bound to the small and high-priced C4+ ion - Explain more

Response: The general rule is that heavy isotopes move into the most tightly bonded phase. To make it clear, we have added some more explanations in lines 291-293 of the revised manuscript with track changes.

Line 258: PEP enzymes resulting… What does PEP mean?

Response: We have accepted these comments. PEP (phosphoenolpyruvate) enzyme is a biological enzyme which exists in C4 plants and plays an important role in photosynthesis. It controls the fractionation of carbon isotope. To make it clear, we have explained more in lines 331-332 of the new manuscript with track changes.

Line 263: indicating that TN contained inorganic nitrogen Not clear, explain more

Response: We have taken this comment seriously. The organic carbon content gradually decreases with increasing profile depth. If not contained inorganic nitrogen, the intercept of ordinate would be zero (b=0). To make it clear, we have explained it in line 339 of the new manuscript with track changes.

Lines 265 to 269: explanations not clear. What is the meaning of organic matter occurred below the middle of A1? Or the middle δ13CTOC became lighter?

Response: This is a nice suggestion that has pointed out what we have ignored before. A1 and A2 are relatively consistent in sedimentary environment. In addition, the variation of material source in A2 gives us reason to speculate that the input of terrestrial organic matter was below the middle of A1. But we have made a mistake in the line 269 about “the middle δ13CTOC became lighter”. We have added more explanation and correct this mistake in lines 342-348 of the new manuscript with track changes.

In Figure 13 the name B1 is missing.

Response: B1 is not missing but blocked by lines. We have moved it to make it more noticeable in Figure 13.

Line 357: the "northern deposition" of Weizhou Island came from the island; from the ocean?

Response: We would like to emphasize that the conclusion we made is that the sediment of Weizhou Island is partly from the island land, not all from the ocean. To make it clear, we have modified in lines 456-457 of new manuscript with track changes.

Lines 143, 150, 257, 297: is relatively positive/negative Replace by positively/negatively biased.

Line 152: the degree of negative bias is weak Replace by the negative bias is…

Line 204: the carbonates carried by marine vapor Replace by carried by marine spray

Response: By taking these suggestions raised by the reviewer, we have replaced these words in lines 183, 192, 193, 265, 330 and 380 of manuscript with track changes.

Reviewer 3 Report

Due to its geochemical approach, this article is interesting and allows to understand the different terrestrial and oceanic interactions leading to coastal sediments of the Weizhou Island. In particular, the authors use the isotopic geochemistry of carbon, nitrogen, and oxygen to trace the origin of the various organic and inorganic inputs.

However, the density of information means that more explanations are needed.

Major revision:

Lines 149-150: the distribution shows a positive correlation with δ13Ccarb. Difficult to interpret, a Figure/correlation coefficient are missing

Line 164: the TN distribution is nearly consistent with that of TOC. Not obvious, needs more explanation.

Line 175: which is roughly positively correlated with δ13CTOC. More explanation is required (what is the correlation coefficient?)

Minor revision:

In the figure 1 you should emphasize the index 1 corresponds to intertidal higher indexes to supra-tidal

Line 13: AMS radiocarbon dating ages (Accelerator Mass Spectroscopy?)

Line 73: sea level decline (explain)

Line 113: AD isotope mass spectrometer (explain AD)

Line 135: sample A1-8 Specify the depth

Line 143: the distribution is proportional to the TIC. What does it mean?

Lines 143, 150, 257, 297: is relatively positive/negative Replace by positively/negatively biased.

Lines 145, 152: anomalies appear at the low parts of A3 and C2 I see nothing

Line 152: the degree of negative bias is weak Replace by the negative bias is…

Why δ13C Is represented on a decreasing scale in Fig. 3 and on an increasing scale in Fig. 7?

Line 186: The stratum was deposited in order from the surface to bottom? It is the contrary!

Line 204: the carbonates carried by marine vapor Replace by carried by marine spray

Line 228: O prefers to be bound to the small and high-priced C4+ ion - Explain more

Line 258: PEP enzymes resulting… What does PEP mean?

Line 263: indicating that TN contained inorganic nitrogen Not clear, explain more

Lines 265 to 269: explanations not clear. What is the meaning of organic matter occurred below the middle of A1? Or the middle δ13CTOC became lighter?

In Figure 13 the name B1 is missing.

Line 357: the "northern deposition" of Weizhou Island came from the island; from the ocean?

Author Response

Response to Review Comments (water-994928)

We are greatly encouraged by the positive comments by the Guest Editor and the Chief Editor in view of our previously submitted manuscript through a peer review. A special thanks to the Guest Editor, the Chief Editor, and three anonymous reviewers for their deep and thorough reviews. We have tried our best to modify the previous submission of this manuscript regarding all the detailed comments and constructive suggestions.

To better show how and where the earlier version is thoroughly revised, both the “track changes” and clean version of the manuscript based on the previous submission are provided. All line numbers mentioned in this detailed reply belong to the manuscript with “track changes”.

Reviewer #3:

Headlights: I suggest writing better the first sentence.

Response: We have taken this comment seriously. We regard a combination of technologies as a point of innovation and have modified in Highlight 1.

Line 21: Can you give a number of this decrease sharply?

Response: By taking this suggestion into account, we have added the drop rate of TIC in in line 25 of the new manuscript with track changes.

Line 28-29: It is very ambiguous the word rapidly   

Response: We have taken this comment seriously. However, due to lack of precise data to constrain this observation, we have deleted the word rapidly in line 33 of the revised manuscript with track changes.

Figure 1: Can you add scale to left panels.

Response: We have added scale to left panels in Figure 1.

Line 73-82: I see here the goal of the manuscript, but you do not explain the novelty of the research, I observe that you asses the same that previous work in other field sites, and you did it in Weizhou Island, where previously has been analyzed through similar processes. I suggest rewriting this paragraph and emphasize it.

Response: Thanks so much for your deep reviews. These constructive comments and detail suggestions have indeed helped us to improve the quality of this manuscript greatly. This suggestion mentioned about previous work in other field sites is for the reason that we would like to refer to it to introduce the background of Weizhou Island in this paragraph. We have rewritten this paragraph and emphasized the novelty of the research by searching for new references. Meanwhile, we have modified other contents including Abstract and Conclusion to improve the quality and logic. See the revision in lines 31-32, 83-110, and 425-427 in the revised manuscript with track changes, respectively.

New references:

  1. Zheng, Z.Y.; Li, G. X.; Tang, C. L.; Zhou, X. Mean sea level changes near Weizhou island from 1969 to 2010. J. OCEAN UNIV., 2014. 13, 369-374.
  2. Ning, Z.M.; Yu, K. F.; Wang, Y. H.; Huang, X. Y.; Han, M. W.; Zhang, J. Carbon and nutrient dynamics of permeable carbonate and silicate sands adjacent to coral reefs around Weizhou Island in the northern South China Sea. Estuar. Coast. Shelf Sci., 2019, 225, 9.
  3. Xu, S.D.; Yu, K. F.; Zhang, Z. N.; Chen, B. A.; Qin, Z. J.; Huang, X. Y.; Jiang, W.; Wang, Y. X.; Wang, Y. H. Intergeneric differences in trophic status of Scleractinian Corals from Weizhou Island, northern South China Sea: implication for their different environmental stress tolerance. J. Geophys. Res.-Biogeosci., 2020. 125, 14.

Subsection Sample Collection: The sample are collected in the same day, in which interval? There is not influenced by spring/neap tide?

Line 144-145: So if it is this influence of the intertidal/supertidal area you have to collect data from different interval to represent every possibility.

Subsection 4.1.2.: If it is true, you have to collect data in different interval too. For example in strong wind, strong wave…Maybe you have to characterized the wave climate to understand it.

Response: We have taken these comments seriously. Most of samples collect during low tide in the same day. To make it clear, we have added exact date and completed the collection information in line 117-118 of the new manuscript with track changes.

Subsection 3.2 and 3.3: I suggest explaining better the figures, and trying to conclude the principal results.

Response: We have taken this comment seriously. To make it better, we have modified and tried to conclude the principal results. See the revision in lines 202-204, 210-211, 220-223, 229 and 236 in the revised manuscript with track changes, respectively.

Section of conclusion must be improved. It is weak. Here also, you have to show the novelty of the manuscript to the community of researchers.

Response: By taking this suggestion into account, we have added the information about the dominant vegetation type in conclusion and have shown the novelty of manuscript to the community of researchers. See the detail revision in Conclusion and lines 403-409 and 425-427 in the revised manuscript with track changes, respectively.

The manuscript is presented correctly, but it should emphasize the novelty of it and the contribution to the scientific community. In addition, the sample collection section should be better explained indicating in which moments the samples have been taken, since the study can only be centered for a certain moment without taking into account the tidal or ocean dependence.

Response: Thanks so much for your deep reviews. These constructive comments and detail suggestions have indeed helped us to improve the quality of this manuscript greatly. Firstly, we have added more contents to improve the novelty of this paper, i.e., (1) adding more information about previous research on Weizhou Island; (2) emphasizing what the novelty and contribution is to the scientific community. See the details revision in lines 83-110. In addition, we have added exact date and have completed the collection information in lines 117-118 of manuscript with track changes. Most of samples collect during low tide in one day. Therefore, we do not take into account the influence of interval.

Round 2

Reviewer 2 Report

I have to admit that the manuscript has improved in quality and presentation of the results. Furthermore, a great effort has been made to explain the novelty of the manuscript as well as the contribution to the scientific community. However, they have to work in the manuscript when they explain the results, they should provide the reader with a comprehensive reading referencing all the figures that are being discussed.

Author Response

Reviewer 2

I have to admit that the manuscript has improved in quality and presentation of the results. Furthermore, a great effort has been made to explain the novelty of the manuscript as well as the contribution to the scientific community. However, they have to work in the manuscript when they explain the results, they should provide the reader with a comprehensive reading referencing all the figures that are being discussed.

Response: We authors are really appreciated for the highly positive comments through your evaluation of our revised manuscript. By taking these suggestions, we have add more detail information and/or have made proper modifications in lines 173, 191, 209, 246, 279, 350, 378, 381, 395, 399 and 415 of revised manuscript with track changes.

Reviewer 3 Report

Corrections have been made. The text is clear, the process interesting and well documented. The research contributes significantly to increasing knowledge about the sedimentation processes on the Weizhou Island.

I propose that the paper be published as is.

Author Response

Reviewer#3

Corrections have been made. The text is clear, the process interesting and well documented. The research contributes significantly to increasing knowledge about the sedimentation processes on the Weizhou Island.

I propose that the paper be published as is.

Response: I would like to convey my heartfelt thanks to you for the thorough evaluation of our revised manuscript. Your positive feedbacks and previous comments have greatly help us to improve the quality and have encouraged us to put forward our research.
